# Understanding the Role of LLMs in Multimodal Evaluation Benchmarks

## Abstract

The rapid advancement of Multimodal Large Language Models (MLLMs) has been accompanied by the development of various benchmarks to evaluate their capabilities. However, the true nature of these evaluations and the extent to which they assess multimodal reasoning versus merely leveraging the underlying Large Language Model (LLM) backbone remain unclear. This paper presents a comprehensive investigation into the role of LLM backbones in MLLM evaluation, focusing on two critical aspects: the degree to which current benchmarks truly assess multimodal reasoning and the influence of LLM prior knowledge on performance. Specifically, we introduce a modified evaluation protocol to disentangle the contributions of the LLM backbone from multimodal integration, and an automatic knowledge identification technique for diagnosing whether LLMs equip the necessary knowledge for corresponding multimodal questions. Our study encompasses four diverse MLLM benchmarks and eight state-of-the-art MLLMs. Key findings reveal that some benchmarks allow high performance even without visual inputs and up to 50% of error rates can be attributed to insufficient world knowledge in the LLM backbone, indicating a heavy reliance on language capabilities. To address knowledge deficiencies, we propose a knowledge augmentation pipeline that achieves significant performance gains, with improvements of up to 60% on certain datasets, resulting in a approximately 4x increase in performance. Our work provides crucial insights into the role of the LLM backbone in MLLMs, and highlights the need for more nuanced benchmarking approaches.

## 1 Introduction

The rapid development of Large Language Models (LLMs) (Touvron et al., 2023; Bai et al., 2023a), combined with advancements in visual encoders (Radford et al., 2021; Zhai et al., 2023) and modality bridge techniques (Liu et al., 2023a; Dai et al., 2023), has catalyzed the evolution of Multimodal Large Language Models (MLLMs) capable of comprehending diverse multi-modal inputs. Concurrently, diverse benchmarks and leaderboards have emerged to evaluate various multimodal perception and reasoning capabilities (Lu et al., 2022b; Lerner et al., 2022; Yue et al., 2024a).

While these benchmarks aim to assess multimodal capabilities, the role of the underlying LLM backbone in MLLM performance remains poorly understood. Recent studies (Tong et al., 2024; Yue et al., 2024c) have highlighted that some benchmarks demonstrate an excessive dependence on the language model component, allowing MLLMs to achieve high scores even without visual inputs. This observation raises critical questions about the true nature of multimodal reasoning in these models and the extent to which performance is driven by the LLM backbone rather than multimodal integration. Furthermore, as different MLLMs utilize LLM backbones with distinct knowledge priors learned from various pre-training corpora (Gao et al., 2020; Penedo et al., 2023), this knowledge inconsistency leads to incomparable evaluation scores when comparing MLLMs with different underlying LLMs. These issues can result in misinterpretation of evaluation scores and may misguide research and deployment of MLLMs by providing an inaccurate assessment of their true multimodal capabilities.

In this paper, we present an in-depth investigation into the role of LLM backbones in MLLM evaluation, focusing on two key aspects: (i) the extent to which current benchmarks truly assess multimodal reasoning versus relying on language capabilities alone, and (ii) the influence of LLM prior knowl-

edge on final performance. To address the first question, we propose an approach that goes beyond simply evaluating models without visual cues (Tong et al., 2024; Chen et al., 2024a). Our method incorporates comparisons with shuffled options and transforms multiple-choice quesiont-answering (QA) formats into open-ended generation tasks, providing a more comprehensive understanding of the role of language capabilities versus true multimodal reasoning in these benchmarks. For the second question, we develop an automatic knowledge identification method utilizing external knowledge bases such as Wikipedia or powerful LLMs (OpenAI, 2023b) to obtain the necessary knowledge behind each question. With these knowledge facts prepared, we examine whether the underlying LLM backbone possesses the requisite world knowledge for multimodal questions, enabling a better understanding of the obtained scores.

We select four benchmarks covering different capabilities of MLLMs: the comprehensive evaluation benchmark MMMU (Yue et al., 2024a), ScienceQA for multimodal reasoning evaluation (Lu et al., 2022b), and two knowledge-based VQA tasks: Viquae (Lerner et al., 2022) and InfoSeek (Chen et al., 2023). Our experimental results with eight MLLMs reveal significant insights into the role of LLM backbones: (i) **LLMs would exploit the shortcuts in question and options, making predictions without relying on the visual inputs.** For example, on the commonly adopted MMMU dataset, accuracy scores remain the same for more than 80% of samples even without visual inputs. Further comparison of datasets and task formats suggests that knowledge-intensive VQA benchmarks requiring entity recognition from images are less affected by this issue, and LLMs could solely rely on options to achieve prediction without relying on visual inputs. Specifically, we observe that the average performance difference between scenarios with and without visual inputs is markedly lower for multiple-choice questions in MMMU (15%) compared to open-ended questions in InfoSeek (65%). (ii) **MLLMs performance shows great dependence on the knowledge of LLM backbones.** We observe that up to 50% of error rates on multimodal benchmarks could be attributed to insufficient world knowledge in the LLM backbone. Besides, MLLMs adopting knowledgeable LLMs such as LLaVA-Next-Yi-34B and InternVL2-Llama3-76B during evaluation tend to perform better, highlighting the significant impact of the LLM backbone on overall performance. Motivated by these findings, we develop a simple knowledge augmentation pipeline to retrieve supplementary background knowledge for answering challenging VQA questions. This approach yields an average significant 36% absolute accuracy gain, with Phi-3 achieving an impressive improvement of over 60% on the Viquae dataset. Further analysis demonstrates a trade-off between knowledge recall and the noise introduced by retrieved knowledge paragraphs.

Our study provides crucial insights into the role of LLM backbones in MLLM evaluation and highlights the need for more nuanced benchmarking approaches that can distinguish between language model capabilities and true multimodal reasoning. These findings have important implications for the development and evaluation of future MLLMs, suggesting that both visual integration techniques and the choice of LLM backbone are critical factors in achieving robust multimodal performance.

## 2 METHOD

In this section, we perform an approach to better understand the role of LLM in multi-modal evaluation benchmarks. Figure 1 provides a comprehensive overview of the method. We begin by formally introducing the key notations essential for setting up our framework (§2.1). Following this, we delve into the specifics of how we measure the significance of vision and knowledge. First we outline the methodologies for evaluating the role of vision (§2.2). We then explore the methodologies for gauging the impact of factual knowledge §2.3 and develop a knowledge-augmented framework to assist the MMLMs (§2.4).

### 2.1 PROBLEM NOTATIONS

VQA involves providing a model with visual input and a related question, and then requiring the model to generate an appropriate answer. Let $D$ be a given multi-modal dataset. For any data entry $d$ in $D$, we define $d$ as a triple $(I, Q, A)$, where $I$ denotes the visual input (a single image in our work), $Q$ represents the textual question, and $A$ is the corresponding answer. We posit that MLLMs process VQA tasks through two primary stages: visual perception and knowledge reasoning. In the visual perception stage, the model extracts key information from the image input. Subsequently, we hypothesize that the model internally reformulates the original VQA question into a cognate

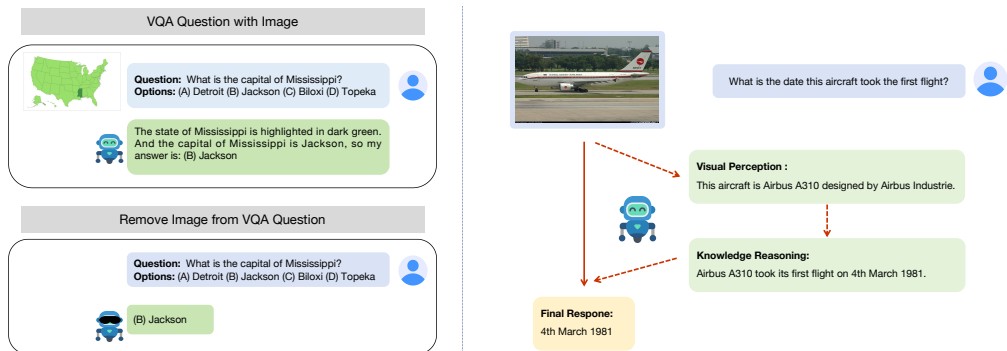

Figure 1: **Left**:We first identify VQA questions answerable without images. **Right**:We subsequently decompose the process of solving visual questions into two distinct yet interrelated steps, decoupling visual perception capability from knowledge.

knowledge reasoning query, denoted as $K$. This query $K$ integrates both the textual input and the extracted visual information, forming the basis for the ensuing reasoning process to derive the answer.

Regarding the representation of the multi-modal large language model (MLLM), we employ the function form $f$, where $f(I, Q), f(\varnothing, Q)$, and $f(\varnothing, K)$ denote the model's responses to visual questions with images, without images, and to knowledge reasoning queries respectively. The visual perception step is described by $P$, with $\sim$ used instead of $=$ to reflect the non-deterministic nature of visual conception. For evaluation, we define a combination $C$ as one of these three input types. Given a dataset $D$, we calculate the Score Rate (SR) as:

$$\text{SR}_D^C = \frac{1}{|D|} \sum_{d \in D} \mathbb{1}\left[f(C_d) == A\right].\tag{1}$$

For instance, $\text{SR}_{Viquae}^{(\varnothing, K)}$ represents the model's performance on knowledge reasoning questions in the Viquae dataset. Empirically, we believe that a higher SR of a model indicates stronger performance, and vice versa.

## 2.2 IS VISUAL CAPABILITY NECESSARY?

Previous research has demonstrated that the absence of visual input often does not significantly impact model performance on certain visual evaluation datasets (Goyal et al., 2017; Tong et al., 2024; Huang et al., 2024). To elucidate this phenomenon, we extend prior work by systematically modifying the VQA task paradigm to assess the role of visual information under varied conditions. Our methodology involves presenting identical questions to models in image-present and image-absent contexts. Furthermore, we introduce two critical modifications to the multiple-choice format: (1) randomization of multiple-choice option order and (2) reformulation of questions into open-ended queries. These alterations serve dual purposes: randomization of options mitigates potential biases towards specific answer types that may align with training data distributions, while open-ended reformulation allows us to evaluate whether the apparent diminished reliance on visual information is an artifact of constrained multiple-choice setups, where some options may be trivially eliminable. Our findings indicate that the presence of multiple-choice options significantly reduces both task difficulty and the necessity for visual information processing. This insight offers a nuanced perspective on the observed similarity in model performance across image-present and image-absent conditions in certain VQA datasets, underscoring the critical role of task design in accurately assessing visual reasoning capabilities.

To quantify these effects, we conduct a comparative analysis of performance differentials between image-present and image-absent scenarios at the dataset level for each model. We also introduce the

Gap Rate (GR) metric, defined as:

$$\text{GR}_D = 1 - \frac{\text{SR}_D^{(\varnothing, Q)}}{\text{SR}_D^{(I,Q)}}, \tag{2}$$

to normalize for inherent variability in model capabilities. Theoretically, a well-constructed multimodal dataset should elicit correct responses predominantly when visual input is provided, with performance in the absence of images approximating chance levels. Consequently, the GR serves as an indicator of a dataset's efficacy in assessing genuine visual reasoning capabilities, with higher values suggesting a stronger coupling between visual information and task performance.

## 2.3 Do MLLMs Have Sufficient Prior Knowledge?

Based on our hypothesis that the resolution of visual tasks could be delineated into two distinct steps, upon receiving an image $I$ and a textual question $Q$, the model implicitly engages in a visual perception process $P$ to generate a corresponding knowledge reasoning problem $K$, then subsequently utilized it to make response. We formalize the entire process as follows:

$$\mathbb{1}\left[f(I,Q) == a\right] = \mathbb{1}\left[P(I,Q) \sim K\right] \cdot \mathbb{1}\left[f(\varnothing, K) == a\right]. \tag{3}$$

The formula suggests that language prior knowledge and visual perception capabilities are equally crucial and indispensable. Therefore, the reason for models' poor evaluation results may not only stem from insufficient visual capabilities but also from a lack of knowledge.

To determine whether the model's knowledge is sufficient, we use knowledge reasoning questions corresponding to visual questions in each dataset as models' inputs and then evaluate their performance. For datasets that do not provide corresponding knowledge reasoning questions, we directly replace image-referenced content in visual questions with given entities or invoke GPT-4[1] (Achiam et al., 2023) to convert the original visual questions (specific prompts employed are detailed in the Appendix A.1).

We also perform a statistical analysis about model's correctness and errors in each visual question and its corresponding knowledge reasoning question. To quantify the analysis results, we introduce the following two indicators, Sufficiency Ratio (SuR) and Necessity Ratio (NeR), defined as follows:

$$\text{SuR}_D = \frac{\sum_{d \in D} \mathbb{1}\left[f(I_d, Q_d) == A_d \mid f(\varnothing, K_d) == A_d\right]}{\sum_{d \in D} \mathbb{1}\left[f(I_d, Q_d) == A_d\right]} \tag{4}$$

$$\text{NeR}_D = \frac{\sum_{d \in D} \mathbb{1}\left[f(I_d, Q_d) \neq A_d \mid f(\varnothing, K_d) \neq A_d\right]}{\sum_{d \in D} \mathbb{1}\left[f(I_d, Q_d) \neq A_d\right]}. \tag{5}$$

These ratios serve to elucidate the sufficiency and necessity relationship between prior knowledge and visual capability, where higher values signify a more robust relationship.

## 2.4 Can Knowledge Augmentation Improve Multimodal Capabilities?

In real-world scenarios, models often encounter the issue of insufficient knowledge due to their smaller scale or outdated information (Gao et al., 2023). To mitigate the limitation caused by the absence of prior knowledge, we adopt a straightforward idea here, using the Retrieval-Augmented Generation (RAG) approach to effectively enhance the model's knowledge and then design proper experiments for effectiveness evaluation (Weston et al., 2018; Cai et al., 2019).

We evaluate the relevance between the knowledge reasoning problem and the paragraph using cosine similarity, and then rank the paragraphs accordingly. Ultimately, the highest-ranked paragraphs are incorporated into the input to enhance the model's knowledge base. Within the framework of RAG, for the top $n$ most relevant paragraphs $p_1, p_2, ..., p_n$ from candidate knowledge document corpus $\mathcal{C}$, we articulate the calculation of Score Rate (SR) as follows:

$$\text{SR}_D^{\text{RAG}_n} = \frac{1}{|D|} \sum_{d \in D} \mathbb{1}\left[f(I, (Q, p_1, p_2, ..., p_n)) == A\right]. \tag{6}$$

Specifically, we employ the state-of-the-art embedding model, NV-Embed-v2 (Lee et al., 2024; Moreira et al., 2024) as our retriever. Following the calculation of similarity, we select 1, 3, 5, and 10 as values for $n$ and evaluate the performance against the vanilla setup.

---

[1]We use the `GPT-4o-2024-05-13` version.

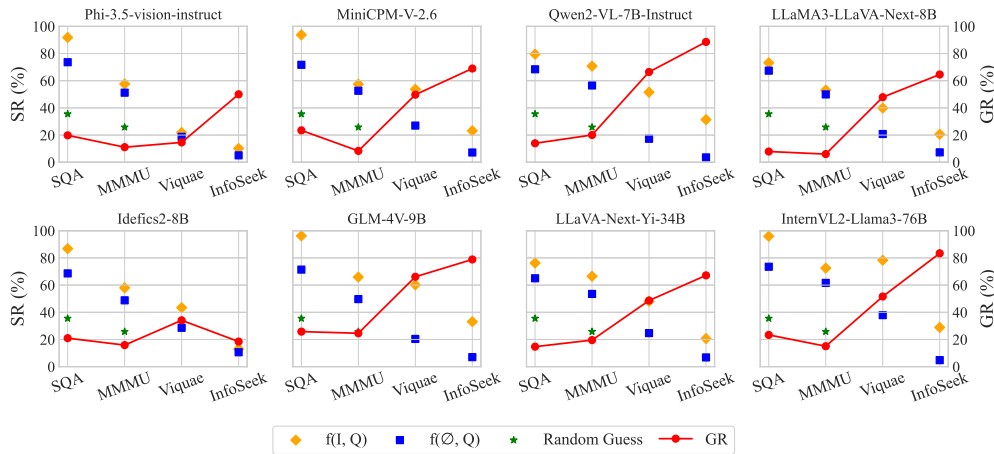

Figure 2: SR comparison of MLLMs under both image-present and image-absent conditions across four benchmarks. $f(\varnothing, Q)$ is relatively close to $f(I, Q)$, but far from Random Guess, indicating that the model's utilization of visual information is low.

## 3 EXPERIMENTS

As described in the preceding sections, we conduct extensive experiments to investigate the role of LLMs in MLLM evaluation. We first introduce the experimental settings (§3.1. We then discuss our findings regarding the shortcuts used by LLMs during evaluation (§3.2) and the knowledge deficiency (§3.3). Finally, we illustrate interesting cases during our investigation (§3.4) and evaluate the effectiveness of our knowledge augmented method (§2.4).

### 3.1 EXPERIMENTAL SETUP

**Benchmarks.** The datasets we use primarily assess the models' knowledge, generalizability, and reasoning abilities, and do not include datasets that primarily rely on visual recognition capabilities such as OCR. The specifics are as follows:

- **Viquae** (Lerner et al., 2022): Viquae comprises a test set of 1257 questions, is a visual version of the Named Entity Question Answering dataset, requiring the identification of named entities in images and then reasoning to answer questions based on the model's inherent knowledge.
- **ScienceQA** (Lu et al., 2022a): SQA consists of multimodal multiple-choice questions on various topics which is sourced from elementary and secondary school science curricula. We selected questions from the test set that have visual context for testing, totaling 2017 items.
- **InfoSeek** (Chen et al., 2023): InfoSeek is a dataset composed of visual information-seeking questions necessitating the model to draw upon fine-grained knowledge learned from pretraining instead of commonsense knowledge to formulate responses. We sampled 3000 questions from its validation set for testing purposes.
- **MMMU** (Yue et al., 2024b): MMMU is composed of multimodal questions collected from university exams, quizzes, and textbooks, requiring the model to possess university-level subject knowledge and excellent reasoning abilities. We selected 648 single-image questions from a subset of its validation set for testing (further details of selected subset are available in Appendix B.1).

**Test Models and Setup.** We conduct experiments using open-source multi-modal large models from different sources, ranging in scale from 4.2 billion to 76 billion parameters, including Qwen-VL (Qwen2-VL-7B-Instruct) (Bai et al., 2023b), Idefics (Idefics2-8B) (Laurençon et al., 2024), LLaVA (LLaMA3-LLaVA-Next-8B, LLaVA-Next-Yi-34B) (Li et al., 2024), Phi-3 (Phi-3.5-vision-instruct) (Abdin et al., 2024), ChatGLM (GLM-4V-9B) (GLM et al., 2024), InternVL (InternVL2-Llama3-76B) (Chen et al., 2024b) and MiniCPM (MiniCPM-V-2.6) (Yao et al., 2024). We use a temperature of 0 for all models for deterministic results. To ensure more accurate evaluation results, we employed different evaluation methods for

Table 1: $SR^{(I,Q)}$ and GR of MMMU in different question formats. Higher GR signifies greater utilization of visual information by models in open-ended visual tasks.

| | Original Option | | Shuffled Option | | Open-ended QA | |
|---|---|---|---|---|---|---|
| | $SR^{(I,Q)}$ | GR | $SR^{(I,Q)}$ | GR | $SR^{(I,Q)}$ | GR |
| Phi-3.5-vision-instruct | 57.6 | 11.0 | 51.9 | 6.0 | 7.1 | 10.9 |
| MiniCPM-V-2.6 | 57.4 | 8.3 | 53.5 | 13.8 | 10.3 | 34.3 |
| Qwen2-VL-7B-Instruct | 70.7 | 20.1 | **64.4** | 17.3 | 11.9 | **59.7** |
| LLaMA3-LLaVA-Next-8B | 53.1 | 6.1 | 58.2 | **19.4** | 7.7 | 36.0 |
| Idefics2-8B | 58.0 | 15.9 | 48.8 | 0.6 | 6.8 | 2.3 |
| GLM-4V-9B | 65.9 | **24.6** | 56.3 | 15.9 | 10.0 | 40.0 |
| LLaVA-Next-Yi-34B | 66.5 | 19.5 | 62.0 | 11.9 | 10.2 | 39.4 |
| InternVL2-Llama3-76B | **72.5** | 15.1 | 62.7 | 15.3 | **17.1** | 59.5 |

open-ended questions and multiple-choice questions. On open-ended tasks, we evaluate the correctness of models' responses by determining whether the candidate answers are present in the output of models through rule-matching. As to multiple-choice questions, we use DeepSeek-AI (2024) to assess whether the model's reasoning results are correct. The specific prompt used for determining the correctness of the model's outputs can be found in Appendix A.2

**Prompts.** For open-ended problems from Viquae and InfoSeek, we simply use the questions as input into the models. For multiple-choice questions from ScienceQA and MMMU, we concatenate the questions and options as the example shown in the Appendix A.3 to form the model's prompt without appending any additional information such as the topic in MMMU or the hint in SQA.

## 3.2 LLMs Exploit Shortcuts in Vision Tasks

We conduct comparative experiments using eight models on four datasets, comparing the performance of the image-included setup with the image-excluded setup to enhance the broad applicability and reliability of the analysis. We also calculate the expected score for multiple-choice questions via randomly guessing. The outcomes are shown in Figure 2. Nearly all GR values below 0.8 suggest that visual information is not always essential, echoing previous findings (Yue et al., 2024b; Tong et al., 2024). Even on open-ended question-answering tasks like Viquae and InfoSeek, models still achieve average SRs of approximately 0.25 and 0.07 without using visual inputs. This could be due to the model having learned similar data during its training process, as the data for these datasets is sourced from Wikipedia, which is widely used in pre-training or supervised fine-tuning stages. As to multiple-choice questions like SQA and MMMU, models' average GR on these two datasets is only 0.18 and 0.15, respectively. Such low GR values indicate a negligible role of visual input in performance.

On MMMU, we explore the correlation between question setup and the role of vision by shuffling or removing the initial options of each question, with the results visualized in Table 1. Obviously, our changes to the question setup pose greater challenges to the MLLMs, as almost all models' SRs have decreased to some extent. Analyzing from the perspective of GR, shuffling the initial options has a relatively minor overall impact. However, the removal of options leads to a significant increase in the maximum GR, rising from 24.6 to 59.7 percent, representing an over 100% enhancement. Combining the previous results, we believe that vision plays a more significant role in open-ended questions, as LLMs may potentially exploit shortcuts within the provided options to formulate responses.

## 3.3 MLLMs Suffers from LLMs' Knowledge Deficiency

Knowledge deficiency of Large Language Models (LLMs) in Visual Question Answering (VQA) tasks are evident, even for state-of-the-art systems. As demonstrated in Table 2, InternVL, despite being equipped with the powerful LLaMA3-70B, achieves an average SR not exceeding 90% across various datasets. This performance ceiling is even more pronounced in smaller models, which exhibit average SRs of approximately 70% across diverse datasets. These findings suggest that all models used in our experiments face the challenge of inadequate knowledge.

Table 2: SR of knowledge reasoning questions across four datasets. Almost all models encounter the challenge of insufficient knowledge.

|  | Viquae | InfoSeek$_{sample}$ | SQA$_{IMG.}$ | MMMU$_{val.}$ |
|---|---|---|---|---|
| Phi-3.5-vision-instruct | 65.8 | 43.9 | 83.2 | 64.2 |
| MiniCPM-V-2.6 | 82.7 | 48.8 | 84.7 | 63.3 |
| Qwen2-VL-7B-Instruct | 78.6 | 45.7 | 80.8 | 72.2 |
| LLaMA3-LLaVA-Next-8B | 83.5 | 52.6 | 76.9 | 63.9 |
| Idefics2-8B | 86.4 | 55.5 | 78.8 | 59.7 |
| GLM-4V-9B | 80.0 | 53.0 | 83.1 | 63.3 |
| LLaVA-Next-Yi-34B | 91.3 | 57.7 | 79.6 | 69.3 |
| InternVL2-Llama3-76B | **94.7** | **61.6** | **88.2** | **77.2** |

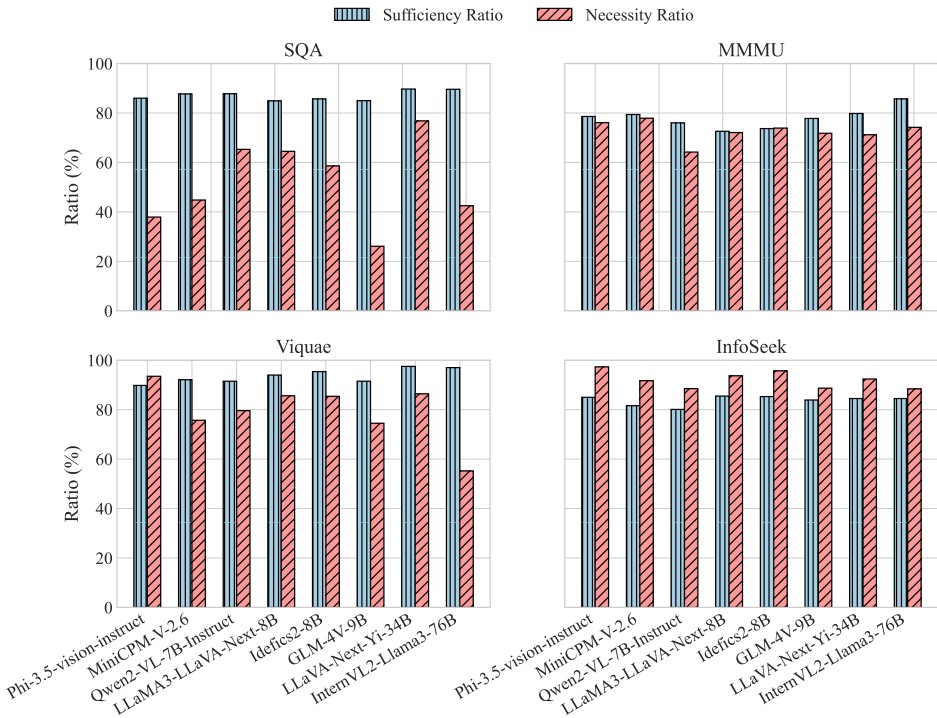

Figure 3: SuR and NeR of different models across four datasets. High values indicate that possessing relevant prior knowledge is a prerequisite for solving visual tasks.

Furthermore, our analysis of SuR and NeR also substantiates the significant impact of prior knowledge on visual capability, as presented in the Figure 3. Taking Phi-3 as an example, it possesses relevant knowledge for over 85% of the visual questions it correctly answered on Viquae. At the same time, over 95% of its knowledge reasoning errors on the InfoSeek dataset are accompanied by failures in their corresponding visual tasks, indicating that knowledge deficiencies are likely a significant factor impairing the model's performance. Our findings suggest that the model heavily relies on relevant knowledge when solving visual tasks, implying the suboptimal performance of MLLMs may stem from the knowledge deficit in their backbone LLMs.

## 3.4 CASE STUDY

We present specific cases in Figure 4 that challenge our initial assumption regarding the decomposition of visual tasks into perception and reasoning steps. An example involves a question about the venue of The Beatles' last ever live concert. In the VQA context, the model correctly identifies The Beatles in the image and subsequently deduces that Candlestick Park was the venue for their last

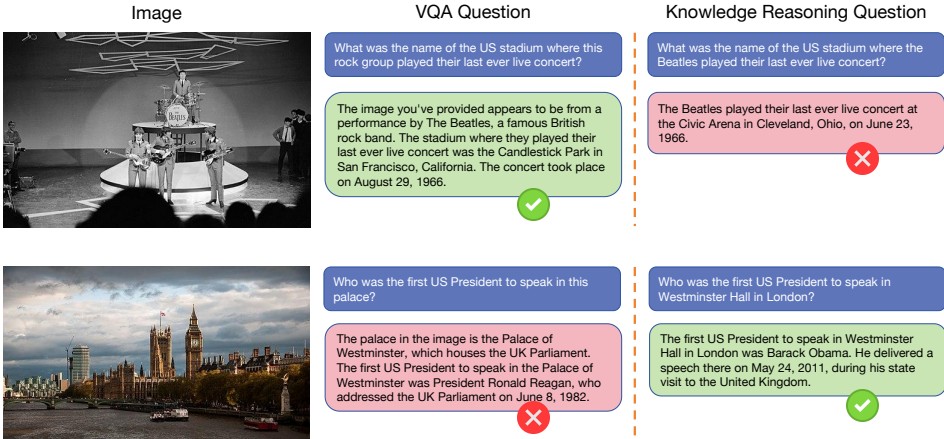

Figure 4: **Top**: While successfully answering the visual question, the model fail to perform well on knowledge reasoning tasks. **Bottom**: The model has relevant knowledge but exhibits hallucinations when addressing visual questions.

Table 3: Recall of the embedding model on knowledge-intensive VQA datasets.

|  | Recall@1 | Recall@3 | Recall@5 | Recall@10 | Recall@50 |
|---|---|---|---|---|---|
| **Viquae** | 45.2 | 65.0 | 73.3 | 82.3 | 91.7 |
| **InfoSeek** | 77.9 | 91.3 | 94.1 | 96.2 | 97.9 |

concert. Paradoxically, when presented with the same query as a pure knowledge reasoning question without visual input, the model fails to provide the correct answer. This observed performance disparity may be attributed to the knowledge representation within the model's architecture. We hypothesize that the relevant information is encoded in the model's parameters in a manner that is more closely aligned with visual question-answering paradigms. Consequently, the presence of this image in the input potentially serves as a more effective retrieval cue, facilitating the model's access to pertinent knowledge.

The model sometimes demonstrates proficiency in accurately answering knowledge reasoning queries, exemplified by its correct responses regarding Barack Obama. However, when confronted with visual questions, it exhibits a propensity for hallucination during the reasoning process, despite accurately identifying the Westminster Hall. This discrepancy suggests a misalignment between visual and textual modalities. While the model possesses the requisite knowledge, as evidenced by its performance on purely text-based queries, it struggles to effectively apply this prior knowledge to visual task resolution.

### 3.5 RETRIEVED KNOWLEDGE BOOSTS MULTIMODAL ABILITIES

Since the lack of knowledge is inevitable, to compensate for this deficiency, it is natural to consider using the RAG approach to enhance the model's knowledge. We employ the embedding model to retrieve the most relevant content from Wikipedia (`June 2024 Wikipedia dump`) for each knowledge reasoning question on InfoSeek and Viquae, and incorporate the information into the corresponding input. The recall on this two datasets is presented in Table 3. The performance of all models in solving visual tasks has significantly improved after knowledge enhancement, as shown in the Figure 5. Nevertheless, in contrast to the recall that increases with the number of relevant documents, the model's SR demonstrates a trend of initially rising and then slightly decreasing., which may be attributed to the noise introduced by an excessive number of relevant paragraphs. In summary, supplementing knowledge significantly enhances the model's performance on visual tasks, which can be applied to model evaluation to minimize differences in relevant knowledge and focus more on visual capabilities.

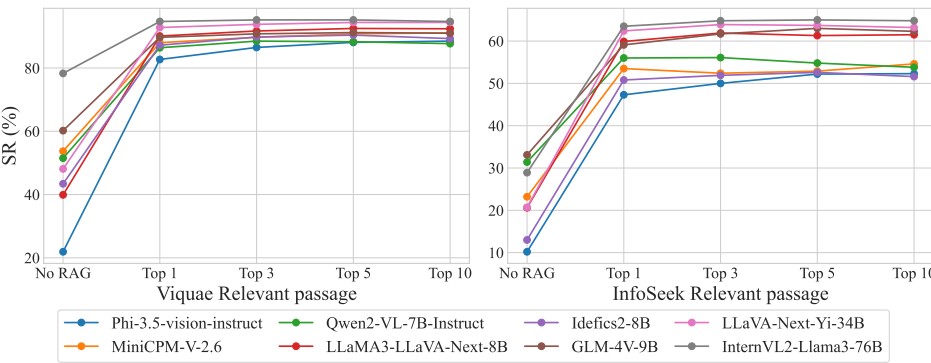

Figure 5: Differences in SR between scenarios without Retrieval-Augmented Generation (RAG) and those using RAG with 1, 3, 5, and 10 relevant documents. Knowledge enhancement significantly improves model performance.

## 4 RELATED WORK

**Multimodal Large Language Models.** Multi-modal Large Language Models (MLLMs) have made remarkable strides in recent years (OpenAI, 2023a; Reid et al., 2024; Ormazabal et al., 2024), demonstrating an unprecedented ability to understand and generate content that seamlessly integrates visual and textual information (Fu et al., 2023). Representative proprietary commercial models, such as OpenAI's GPT-4o (Achiam et al., 2023), Google's Gemini 1.5 Pro (Reid et al., 2024), and Anthropic's Claude 3.5 Sonnet (Anthropic, 2024), have showcased impressive capabilities in various tasks. On the open-source front, models like LLaVA (Liu et al., 2023a), Qwen-VL (Bai et al., 2023b) and Phi-Vision (Abdin et al., 2024), have also demonstrated remarkable progress, particularly in their ability to comprehend multiple images or video simultaneously, expanding the scope of MLLMs from static single images to dynamic multi-frame visual content. Our research aims to gain a deeper understanding of MLLM's performance and limitations, with a special focus on the role of the LLM backbone. Our experimental results show that current MLLMs rely on the LLM backbone heavily on certain benchmarks, and suffer from knowledge deficiency when facing VQA tasks demanding rich world knowledge. Based on our findings, we introduce a RAG-based method that significantly enhances model performance.

**Multimodal Understanding Benchmarks.** The rapid advancement of MLLMs has spurred the development of diverse evaluation benchmarks. These range from specialized tasks like OCR (e.g., InfogrpahicsVQA (Mathew et al., 2022), ChartVQA (Masry et al., 2022), DocVQA (Mathew et al., 2021)), knowledge integration (e.g., Viquae (Lerner et al., 2022) and Infoseek (Chen et al., 2023)), and mathematical reasoning (e.g., ScienceQA (Lu et al., 2022b) and MathVista (Lu et al., 2023)), to comprehensive frameworks such as MMMU (Yue et al., 2024b), MME (Fu et al., 2023), MM-Bench (Liu et al., 2023b), and MMVet (Yu et al., 2023). Our work contributes to this landscape by critically examining these evaluation frameworks, echoing previous findings that visual inputs may contribute less significantly in these benchmarks (Yue et al., 2024c; Chen et al., 2024a; Tong et al., 2024). Additionally, we show that the multiple-choice format could become a shortcut that the LLMs could leverage to bypass the visual inputs and we also identify certain errors primarily stem from language knowledge limitations rather than visual perception deficiencies. These findings provide valuable insights for developing more robust evaluation benchmarks, emphasizing the need to disentangle language model capabilities from true multimodal reasoning in MLLM assessment.

## 5 CONCLUSION

This study provides a comprehensive analysis of the role of LLM backbones in Multimodal Large Language Model (MLLM) evaluation, shedding light on critical aspects that have been largely overlooked in previous research. Our investigation reveals several key insights that have significant implications for the development and evaluation of MLLMs. Our experimental findings first show

that LLMs could exploit shortcuts by relying on inappropriate options in visual tasks, and that open-ended questions could offer more robust assessments. Secondly, we identify substantial knowledge deficiencies across various datasets, where models fail to provide correct answers despite accurate visual perception. To mitigate this, we implement a Retrieval-Augmented Generation (RAG) approach, which significantly improved performance on visual tasks by enhancing the models' factual knowledge. Further analysis reveals a phenomenon of knowledge misalignment between visual and textual modalities.

## LIMITATIONS

Since only a portion of the models used in our experiments support multi-image input, and some questions are difficult to accurately convert into corresponding knowledge inference tasks, we selected only a subset of the MMMU dataset. Additionally, due to the scarcity of multi-modal embedding models, we opted to use knowledge inference questions instead of visual questions for retrieval during the RAG process. In future work, we plan to employ multi-modal retrievers to identify the most relevant paragraphs for VQA questions and evaluate the effectiveness.

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

# A PROMPT

## A.1 CONVERT VQA QUESTIONS

Since some datasets do not provide knowledge reasoning questions corresponding to visual questions, we have designed a sophisticated prompt that inputs the original visual input, text question, and corresponding answer into GPT-4 to transform the question. The specific prompt is as follows:

## A.2 MODEL EVALUATION ON MULTIPLE-CHOICE QUESTIONS

Due to the possibility that the model may generate a lot of thinking during the answering process, and the corresponding letters for options such as 'A' or 'B' are likely to appear within the output, the rule-matching method may not be accurate enough. Therefore, we use DeepSeek to evaluate the model's output, resulting in a more accurate assessment. The specific prompt is as follows:

## A.3 INPUT TEMPLATE FOR MULTIPLE-CHOICE QUESTIONS

We simply add corresponding prefixes to the question part and the option part. We also insert a prompt "Answer" at the end of the question to instruct the model to respond to the question. Here is an example from MMMU dataset.

702
703
704
705
706
707
708
709
710
711
712
713
714
715
716
717
718
719
720

You will receive a VQA question and its corresponding answer, as well as the options for the question.
Now based on the provided information, you need to convert the given VQA problem into a textual question by adding image decription to the original question so that blind people can also answer it.
When describing the image, you should focus on the key features and important details that are relevant to the question or help to solve the problem.
Here is the VQA problem:
Question:
  visual question
Options:
  options
The Answer of this vqa problem is:
  ground truth
Options should not be included in the question.
Again, You are describing this VQA question to a blind person, ensuring not to overlook any visual details relevant to the question.
Now, please convert the VQA problem into a textual question. You can think step by step.
The result should be in a **dict** with key "question" and value as the textual question, output format should be:
  {'question':  'your output'}

721
722
723
724
725
726
727
728
729
730
731
732
733
734

You will get a prediction and an answer of the same question, please judge whether the prediction is correct or not.
The answer is two parts, one part is an alphabet, one part is a sentence.
If the prediction can match one part of the answer, then the prediction is correct.
If the prediction can't match any part of the answer, then the prediction is wrong.
Prediction:
  model's response
Answer:
  ground truth
Only output the result, no need to explain, result should be one word "Yes" or "No".
Result:

735
736
737
738
739
740
741
742
743
744
745
746
747
748
749
750
751
752
753
754
755

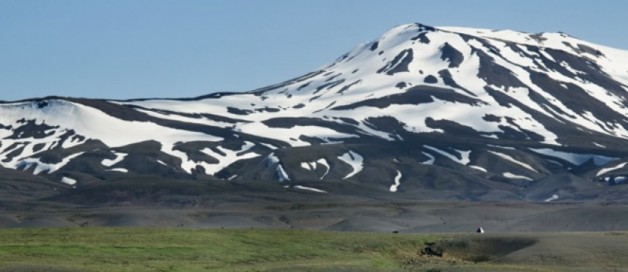

User     **Question**: Identify the biome shown in **IMAGE**
         **Options**:
          (A) taiga
          (B) tundra
          (C) rain forest
          (D) desert
         **Answer**:

| Type | Num. | GPT-4 |
|---|---|---|
| Accounting | 30 | 56.7 |
| Agriculture | 29 | 69.0 |
| Art | 30 | 73.3 |
| Art Theory | 25 | 88.0 |
| Basic Medical Science | 28 | 85.7 |
| Biology | 27 | 51.9 |
| Chemistry | 18 | 61.1 |
| Clinical Medicine | 29 | 89.7 |
| Computer Science | 25 | 68.0 |
| Design | 30 | 80.0 |
| Diagnostics and Laboratory Medicine | 29 | 65.5 |
| Economics | 27 | 81.5 |
| Finance | 22 | 68.2 |
| Geography | 26 | 53.9 |
| History | 28 | 75.0 |
| Literature | 29 | 89.7 |
| Manage | 24 | 66.7 |
| Marketing | 29 | 82.8 |
| Math | 26 | 65.4 |
| Pharmacy | 24 | 83.3 |
| Physics | 28 | 71.4 |
| Psychology | 25 | 80.0 |
| Public Health | 30 | 83.3 |
| Sociology | 28 | 71.4 |

# B   DATASET SETUP

## B.1   MMMU SUBSET

In the table below, we present the specific subsets of MMMU that we selected, along with the number of questions in each subset. Moreover, we provide the Success Rate (SR) using GPT-4 on its transformed knowledge reasoning questions.

