# OpenReview forum: "Understanding the Role of LLMs in Multimodal Evaluation Benchmarks"
_ICLR.cc/2025/Conference — Submitted to ICLR 2025_

### Official Review · Reviewer_WivR · 2024-10-28

**Soundness:** 2
**Presentation:** 3
**Contribution:** 2
**Rating:** 5
**Confidence:** 3

**Summary:**

The paper studies the contribution of language models in multi-modal evaluation benchmarks. The authors aim to quantify the importance of language in the performance on such datasets in two ways: (i) quantify shortcut reasoning (i.e., how much these models look at the image/how many of these questions do not require image understanding), and (ii) quantify the role of prior knowledge in answering these questions (i.e., understand whether more knowledgeable backbones provides better performance). The authors select four benchmark datasets and eight models to investigate these two aspects. As for prior works, they confirm that language models tend to exploit shortcuts, i.e., do not always use visual information to answer. Moreover, they discovered that the performance on the selected benchmarks heavily depends on prior knowledge of the language model.

**Strengths:**

- The work is well-presented and motivated, and the writing is clear.
- The experiments cover four multi-modal datasets and eight multimodal language models.
- The authors introduced some changes to previous evaluation protocols (i.e., options shuffling, open-ended conversion) to minimize biases.
- The two main discoveries present valuable insights into how to design new benchmarks and what to focus on when training multi-modal language models.

**Weaknesses:**

- The analysis of visual capability necessity was already studied in the literature (as also noted by the authors), leaving only the second discovery (i.e., more language knowledge implies better multimodal performance) as the main discovery of the work. Moreover, one could argue that this second point is somewhat intuitive and is more of a confirmation of it.
- The work acknowledges using only a subset of the MMMU benchmark due to limitations in the models' support for multi-image input and the complexity of converting certain questions into corresponding knowledge reasoning tasks.
- Experiments regarding different retrieval approaches are missing, e.g., using different retrieval approaches and potentially exploiting multi-modal information.

**Questions:**

- As I am not fully convinced of the novelty of this work, I would like to ask the authors how their discoveries regarding the necessity of visual information differ from prior works.
- Moreover, I would like to ask the authors whether one could infer the second discovery by evaluating the performance of different pre-trained language models with vision capabilities at different scales (e.g., LLAMA 7B, 30B, etc) without relying on retrieval.
- I expect that adding vision to a language model after pretraining shouldn't diminish its existing knowledge. This means we could evaluate the sufficiency of prior language knowledge for multimodal tasks as if they were language-only tasks. I ask the authors to discuss this and include tables linking knowledge sufficiency results from multimodal benchmarks to language model performance on language tasks, i.e., for models adding vision post-pretraining, report their performance on language benchmarks, and check for correlations between them and multimodal knowledge sufficiency.

---

> ### Author Response · Authors · 2024-12-02
>
> We sincerely appreciate your valuable comments. We found them extremely helpful in improving our manuscript. We address them in detail below.
> > **Weakness 1: The main discovery of the work is somewhat intuitive.**
>
> Thank you for this point. Our focus is more on the role that LLMs play in solving multimodal problems. For the first point, in addition to reiterating previous conclusions, we also point out that LLMs are adept at using options to find shortcuts to solving multimodal problems. For the second point, while we acknowledge that this is quite intuitive, we hope to have a quantitative analysis and attempt to minimize the knowledge differences between different models by using the RAG tool, thereby making a fairer comparison of the visual capabilities of the models.
> > **Weakness 2: Only a subset of MMMU is used.**
>
> Thank you for this point. For certain questions (such as Music and Electronic), it is indeed challenging to automatically convert multimodal problems into corresponding knowledge-reasoning questions. The subset we ultimately selected is over 70% of the full set, so the experimental results should to some extent avoid randomness and are credible.
> > **Question1: What is the new discovery regarding the necessity of visual information?**
>
> Thank you for the question. We focus on how LLMs solve multimodal problems using only textual information. Through comparative experiments with different question formats (multiple-choice vs. open-ended questions), we believe that inappropriate option settings make the questions easier, reducing the necessity of visual input and allowing LLMs to find shortcuts to solving multimodal problems through text alone.
> > **Question2: Is retrieval crucial for inferring the second discovery?**
>
> Thank you for the question. The results obtained from evaluating models of different scales indeed support the second discovery, as shown in Table 2. However, we hope to fairly assess the visual capabilities of models under the existing testing conditions by supplementing their knowledge. Therefore, we chose to use the RAG approach not only to demonstrate the deficiency of knowledge but also to explore its effectiveness in knowledge supplementation.

---

### Official Review · Reviewer_8YJb · 2024-10-30

**Soundness:** 3
**Presentation:** 3
**Contribution:** 2
**Rating:** 5
**Confidence:** 4

**Summary:**

This paper investigates the role of LLM backbones for MLLM in the perspective of which current benchmarks truly evaluate multimodal reasoning and the effect of LLM prior knowledge.  To achieve this, this paper introduces a modified evaluation protocol to elucidate the contribution of the LLM's backbone and an automatic knowledge identification technique to diagnose whether the LLM has the necessary knowledge for the question. This paper reveals that some benchmarks allow high performance without image inputs and many error rates can be attributed to insufficient LLM capabilities. Furthermore, this paper proposes a RAG pipeline for knowledge-based VQA tasks.

**Strengths:**

- S1. The analysis of the contribution of the Language side in MLLM is of great community interest and is one of the key topics


- S2. Incorporating metrics such as SR, GR, SuRD, and NeRD enables more detailed analysis. It will be an important analytical tool.

**Weaknesses:**

- W1. The author's investigation into image-agnostic questions within MLLM benchmarks confirms findings from previous research [1]. Although this analysis provides a more in-depth view, the results that some benchmarks allow high performance without image inputs are not unexpected. Similarly, it is widely recognized that an MLLM’s performance depends on LLM knowledge capacity and generally improves with LLM’s size. Therefore, the finding that “many error rates can be attributed to insufficient LLM capabilities” is not surprising.


- W2. It would be beneficial to propose how to construct the image-dependent questions. For example, this paper could create a benchmark from existing datasets that requires the use of visual information to answer the questions correctly. It would be more beneficial to the community if such a benchmark could be created along with Analysis.

- W3. The evaluation dataset is not new. I would like to know how the proposed metric works on a more recent dataset such as MMStar [1].  MMStar consists of image-dependent questions. Therefore if new issues are identified using the analysis tools in this study, new findings can be given in terms of how to create image-dependent problems.


- W4. The solution for the LLM's knowledge lacking is also just using RAGs, so there is limited novelty.


[1] Chen+, Are We on the Right Way for Evaluating Large Vision-Language Models?, NeurIPS2024

**Questions:**

The role of LLM in MLLM is a very important topic, and this paper provides an analytical tool on that topic. On the other hand, as noted in Weakness, what this paper examines is merely a more comprehensive examination of what is already known, and the findings are not surprising.

I would like to know why existing evaluations are not enough and what are the novel findings through this analysis. And, also, I would like to know what would the results be if the author used newer benchmarks for evaluation.

**Details Of Ethics Concerns:**

Not ethics concern

---

> ### Author Response · Authors · 2024-12-02
>
> We sincerely appreciate your valuable comments. We found them extremely helpful in improving our manuscript. We address them in detail below.
> > **Weakness 1: Findings are not surprising.**
>
> Thank you for this point. We acknowledge that our findings are not entirely unexpected, but we have conducted more detailed analyses of the relevant phenomena through extensive experiments. Additionally, we have focused more attention on the role of LLMs in solving multimodal problems, and we try to align the role of LLMs during the evaluation phase of multimodal capabilities through our discovery, thereby making the evaluation of visual ability more fair.
> > **Weakness 2 : No solution provided for how to construct image-dependent questions.**
>
> Thank you for this point. Regarding how to more fairly test visual abilities, many works have proposed more vision-centric datasets. We, however, analyze and attempt to address this issue from the perspective of the role of LLMs. First, we can filter out questions that do not require visual input, and then by supplementing knowledge, we can minimize the differences between LLMs, thereby aligning the language foundations of various MLLMs and achieving a fair test of visual abilities.
> > **Question1: Why existing evaluations are not enough and what are the novel findings through this analysis?**
>
> Thank you for the question. We point out that LLMs are adept at using the information in the options to find shortcuts that allow them to solve multimodal problems using only textual queries, and that the open-ended question format is more suitable for testing multimodal capabilities.

---

### Official Review · Reviewer_bsD9 · 2024-11-01

**Soundness:** 3
**Presentation:** 3
**Contribution:** 2
**Rating:** 3
**Confidence:** 4

**Summary:**

The paper explores the impact of LLM backbones on the evaluation of MLLMs, concentrating on two primary aspects: the extent to which existing benchmarks accurately measure multimodal reasoning, and the effect of LLMs' prior knowledge on performance. The paper introduces several metrics to disentangle visual perception capabilities from language prior knowledge. The study is conducted on four public MLLM benchmarks and eight open-source MLLM models. Furthermore, the authors introduce a RAG pipeline to enhance the performance.

**Strengths:**

+ Rethinking the impact of the LLM backbone in MLLM evaluation is insightful and may advance the development of MLLM.
+ The paper is generally easy to read and follow.
+ The introduced metrics and RAG pipeline are sound.

**Weaknesses:**

- According to lines 52-55, the reviewer understands that the author aims to focus on two aspects: the influence of language knowledge on the benchmark and the MLLM models, respectively. However, the metrics introduced are all dependent on specific MLLM models, making it impossible to objectively assess the language effects on the benchmark itself. The reviewer suggests that it would be more effective to use ground-truth (human annotation) to address the authors' first question: to what extent do current benchmarks truly assess multimodal reasoning versus relying solely on language capabilities?

- While it is important to disentangle the language prior from MLLM evaluation, the task has already been considered in the construction of existing MLLM benchmarks, for example, SEED-Bench [A] (Sec. 3.3, "... For verifying question/answer pairs, we filter out questions that can be answered correctly by multiple LLMs without resorting to visual information ..."). However, such works have not been discussed in the paper.
[A] SEED-Bench: Benchmarking Multimodal Large Language Models. CVPR 2024.

- The reviewer acknowledges the empirical contributions of this work. However, it offers limited new knowledge to the community. Specifically, the impact of language prior on MLLM evaluation has already been addressed in existing benchmarks (see Weaknesses #2). Additionally, the introduced metrics rely on specific MLLM models without incorporating new ground-truth annotations (see Weaknesses #1), and the RAG pipeline presented is an established technique. The quality of the paper would be significantly enhanced if the authors could reconstruct existing popular benchmarks to disentangle them from language prior and place greater emphasis on multimodal reasoning.

**Questions:**

Please see the Weaknesses.

---

### Official Review · Reviewer_sMeu · 2024-11-03

**Soundness:** 2
**Presentation:** 1
**Contribution:** 1
**Rating:** 5
**Confidence:** 4

**Summary:**

The paper proposes an examination of issues within multi-modal evaluation from two aspects:

	1.	To what extent answers rely solely on textual information
	2.	How much external knowledge can boost the performance of multimodal questions

The paper finds that in some current benchmarks, up to 50% of questions can be solved using only textual queries. The authors introduce a retrieval-augmented generation (RAG) method to improve the language model’s performance.

**Strengths:**

The idea is intuitive and easy to understand.

**Weaknesses:**

* The issue that multimodal questions can often be solved with textual queries alone is not new; many prior works either study or are motivated by this observation (e.g., [1,2]). The paper lacks a more extensive discussion of related works that have studied this phenomenon.
* Equation 3 is confusing. From an intuitive standpoint, it doesn’t seem to hold because: 1) even without the correct knowledge, the model may still provide the correct answer, and 2) in many cases, explicit knowledge is not required (e.g., inferring relative distances between two objects in an image), so such decomposition may not apply.
* It's expected that additional information retrieved by RAG should boost the performance. It would be beneficial to discuss when RAG provides the most advantage. For instance, what types of VQA questions benefit most from RAG, and under what conditions does it perform best?
* It would be nice to provide some results on the OK-VQA dataset as well, as it also requires external knowledge which should benefit from rag.

References:
[1] Fu, X., Hu, Y., Li, B., Feng, Y., Wang, H., Lin, X., … & Krishna, R. (2024). Blink: Multimodal large language models can see but not perceive. arXiv preprint arXiv:2404.12390.

[2] Zhang, J., Mishra, A., Patwardhan, S., & Agarwal, S. (2022). Can open-domain question answering systems answer visual knowledge questions? arXiv preprint arXiv:2202.04306.

[3] Schwenk, D., Khandelwal, A., Clark, C., Marino, K., & Mottaghi, R. (2022, October). A-okvqa: A benchmark for visual question answering using world knowledge. In European Conference on Computer Vision (pp. 146-162). Cham: Springer Nature Switzerland.

**Questions:**

1. In Section 3.3, Table 2, how do you conclude that the performance cap is due to insufficient knowledge? Could it not also be attributed to limited reasoning capabilities, among other factors?
2. Additionally, the writing and notation used in the paper are unclear and need improvement, e.g.
* The connection between some proposed concepts is unclear. There is a missing link between the sections “IS VISUAL CAPABILITY NECESSARY?” and “DO MLLMS HAVE SUFFICIENT PRIOR KNOWLEDGE.”
* The indicator function $\mathbb{1}\left[ P(I, Q) \sim K \right]$ lacks a clear definition of its true conditions. The notation is poorly defined, as  $P(I, Q) \sim K$  is usually interpreted as drawing  K  from the joint distribution of  I  and  Q.

---

> ### Author Response · Authors · 2024-12-02
>
> We sincerely appreciate your valuable comments. We found them extremely helpful in improving our manuscript. We address them in detail below.
> > **Weakness 1: Lacks more extensive discussion.**
>
> Thank you for this point. Indeed, many previous works have mentioned that multimodal questions can be solved with textual queries alone[1, 2]. Some of these works have proposed more vision-centric datasets or testing methods, while our work focuses more on exploring the role of LLMs in this context.
>
> Through Figure 2, we found that in multiple-choice format, models are more likely to answer questions correctly using only textual queries. To compare the impact of different question formats, we chose the well-known benchmark MMMU and converted it into corresponding open-ended question formats without options. The significant increase in GR indicates that the presence of options reduces the necessity of visual input. Additionally, when the order of options was scrambled, SR did not change significantly, suggesting a low likelihood of data contamination on MMMU.
>
> Therefore, we believe that one of the root causes of the issue "multimodal questions can be solved with textual queries alone" is that LLMs are adept at using options to exploit shortcuts to solving problems (Section 3.2).
>
> References:
>
> [1] Tong, S., Brown, E., Wu, P., Woo, S., Middepogu, M., Akula, S. C., ... & Xie, S. (2024). Cambrian-1: A fully open, vision-centric exploration of multimodal llms. arXiv preprint arXiv:2406.16860.
>
> [2] Huang, J., Chen, L., Guo, T., Zeng, F., Zhao, Y., Wu, B., ... & Zhang, M. (2024). Mmevalpro: Calibrating multimodal benchmarks towards trustworthy and efficient evaluation. arXiv preprint arXiv:2407.00468.
> > **Weakness 2: Equation 3 is confusing.**
>
> Thank you for this point. First, answers not based on relevant knowledge are usually based on guessing, which often relies on simpler question settings and is a low-probability event. For the second point, taking "inferring relative distances between two objects in an image" as an example, how to measure the distance is the knowledge required by this problem. Therefore, we believe that this equation is generally in line with the problem-solving process and holds true in most cases.
> > **Question1: How to conclude that the performance cap is due to insufficient knowledge?**
>
> Thank you for the question. Based on our hypothesis, we decompose the solution to visual problems into two stages: visual perception and knowledge reasoning. In the experiments shown in Table 2, we use knowledge reasoning questions without images to prompt different MLLMs, so the performance gap must be unrelated to visual abilities and instead stem from knowledge reasoning abilities. Regarding your mention of limited reasoning capabilities and other factors, these all fall under the part of knowledge deficiency that we have discussed. Essentially, the reasoning process is also about breaking down the problem into various steps and solving them, where how to decompose the problem and how to solve each sub-problem are both manifestations of knowledge.

---

### Meta-Review · Area_Chair_UBZz · 2024-12-20

**Metareview:**

Paper was reviewed by four expert reviewers and received 3 x marginally below the acceptance threshold and 1 x  reject, not good enough ratings. Overall, all reviewers agree that the paper is lacking in a couple of core aspects, mainly: (1) the novelty is marginal, (2) it offers limited new insights to the community (given prior work); and (3) architectural improvements are limited to introducing RAG. Authors have provided rebuttal to address some of the concerns. Reviewers, however, found author responses only partially satisfactory with many concerns being unaddressed. As a result, the rebuttal did little to sway the reviewer opinion.

AC has looked at the reviews, rebuttal, discussion, as well as briefly the paper itself. Overall, AC agrees with the reviewer consensus that paper is not ready for publication in ICLR in its current form. The decision therefore is to recommend Rejection at this time.

**Additional Comments On Reviewer Discussion:**

Authors have provided rebuttal to address some of the reviewer concerns. Reviewers, however, found author responses only partially satisfactory with many concerns being unaddressed. As a result, the rebuttal did little to sway the reviewer opinion.

---

### Decision · Program_Chairs · 2025-01-22

Reject